# ‘Bringing the Covert into the Open’: A Case Study on Technology Appropriation and Continuous Improvement

**DOI:** 10.3390/ijerph19106333

**Published:** 2022-05-23

**Authors:** Michiel Bal, Jos Benders, Lander Vermeerbergen

**Affiliations:** 1Centre for Sociological Research, KU Leuven, 3000 Leuven, Belgium; jos.benders@ntnu.no (J.B.); lander.vermeerbergen@ru.nl (L.V.); 2Department of Industrial Economics and Technology Management, Norwegian University of Science and Technology, 7491 Trondheim, Norway; 3Institute for Management Research, Radboud University Nijmegen, 6525 AJ Nijmegen, The Netherlands

**Keywords:** technology appropriation, continuous improvement, lean, functionalities, work practices, personal digital assistant, nursing

## Abstract

As end-users, employees appropriate technologies. Technology appropriation is generally conceived as a covert phenomenon. In particular, alternative ways and new purposes for which employees deploy technologies tend to remain hidden. Therefore, the potential of technologies as a source of organizational improvements may remain undisclosed. Continuous improvement (CI) programs, in contrast, are explicitly oriented at disclosing organizational improvements. In essence, CI programs encourage employees to openly discuss how to improve their work practices. Such continuous movements towards novel, often better, ways of working may be perfectly suited to bring the covert nature of technology appropriation into the open. Based on a case study on a personal digital assistant (PDA) in a Belgian nursing home with such a CI program in place, we document and analyze to what extent and why functionalities of the PDA were discussed and further developed. We distinguish between the functionalities that, upon implementation, intended to improve particular work practices, and those that surfaced after the technology had been introduced. To conclude, we point at employees’ perceived usefulness of their work practices and their willingness to improve these, rather than only the technology itself, to further the debate on technology appropriation.

## 1. Introduction

Novel technologies continue to be developed and adopted across organizations, aiming to achieve organizational improvements [1,2,3]. However, as technologies are inanimate, it is up to human actors to deploy them in such ways that the intended benefits are actually reached, or even surpassed [2,4]. Applied to the intra-organizational level, this requires gaining insight into the organizational context arguably fostering or hindering how employees appropriate technologies that are in use, or to be used [5]. In essence, technology appropriation, hereafter referred to as ‘appropriation’, may be defined as the way in which employees adopt, adapt and incorporate technologies into their existing working practices [6].

As clearly expressed by scholars studying the ‘situated use of technology’ [5,7,8,9,10], appropriation always takes place within a particular organizational context. Within their organizational context, employees may appropriate technologies in the way technologies ought to be used [10,11,12]. However, employees may also appropriate technologies differently, and for ends that were initially not intended. These types of appropriation are generally covert phenomena [4,6,13]. Something similar applies to workarounds where employees covertly alter their work practices and procedures without approval or awareness of their supervisors and managers [14,15]. Consequently, as with workarounds, employees’ self-initiated changes induced by appropriation may or may not lead to organizational improvements [16].

Reaching such improvements may also be a goal in itself, resulting from deliberate organizational programs that stimulate employees to openly discuss and improve their work practices [17]. Often as a part of implementing ‘lean’, continuous improvement (CI) programs aim to tap into employees’ hands-on knowledge and experience of their own work [18]. The goal of a CI program is actively stimulating employees to come up with suggestions to improve their work practices [19,20]. At regularly organized CI meetings, employees are to discuss how to improve their work practices openly, instead of covertly.

With a CI program in place, the way in which employees adopt, adapt and incorporate technologies into their existing work practices may thus happen out in the open. Discussing technology appropriation is then purposely made overt as employees embed their end-user perspectives in the generic movement towards continuously improved work practices.

It seems evident to study how technology appropriation and CI programs relate. Indeed, it has been suggested that, for instance, workarounds may be undisclosed as part of a CI program: to assess why they came into being, whether they are actually improvements and if not, what alternatives solutions might be viable [14,15]. The same may hold for the appropriation of technologies. Barriers and suggestions to appropriate a technology as intended, as well as for other purposes, become evident in employees’ daily work practices, and may thus be brought into the open as part of a CI program. The question then arises how technology appropriation and CI programs relate. The aim of this study is thus to examine technology appropriation in an organizational context with an improvement program in place. Such a context, that actively fosters employees to discuss their work practices openly, is hypothesized to stimulate employees’ appropriation of technologies as suits them best. This study therefore documents and analyzes technology appropriation in an organization with a CI program in place.

Based on a longitudinal case study in a Belgian nursing home, purposely selected for its CI program and its implementation of a personal digital assistant (PDA), this study’s contributions are as follows. First, a CI program creates opportunities to bring covert appropriation processes into the open. Second, the perceived usefulness of (improved) work practices contributes to understanding why employees appropriate a technology for some purposes, yet not for all.

The remainder of this paper is structured as follows. Firstly, we conceptually draw on technology appropriation (Section 1.1) and continuous improvement (Section 1.2). Secondly, we detail the methods used and interpret why the longitudinal case study suits the aim of this study. Fourthly, we summarize our empirical findings. Lastly, we discuss these findings and provide limitations and future research avenues.

### 1.1. Technology Appropriation

Poole and DeSanctis first broadly defined technology appropriation as ‘the process of employees altering a system as they use it’ [11]. Later, these two authors adapted their definition and changed it to the ‘immediate actions of employees that illustrate deeper structuration processes’ or the rules and resources related to technologies that are typical of an organizational context [12]. Conceivably, they started focusing on scholars studying ‘the situated use of technology’, whose main idea is that employees not merely accept technology use [5,7,9,10,12,21], but instead adopt, adapt and incorporate technology uses into their existing working practices [6,10,12]. However, the extent to which employees either openly or covertly appropriate technologies depends on the organizational context in which the technology is implemented. In turn, rules and resources within an organizational context mediate the extent to which employees can or cannot act with technologies [5,12,22].

As noted earlier, technology appropriation may be undertaken to varying extents and in several ways [6]. It may involve customization in the traditional sense. This means employees may reconfigure the technology as purposely deployed, in order to suit their local needs. However, technology appropriation may also involve employees making use of the technology for its intended purpose but in a different way, as well as for unintended purposes; that is, purposes beyond those for which it was originally implemented. In what follows, we distinguish intended functionalities and unintended functionalities to highlight the functionalities the technology was and was not purposely deployed for, respectively [4,5].

Regarding the appropriation of intended functionalities, employees may follow three avenues. First, employees may appropriate an intended functionality ‘faithfully’, that is, in line with how the technology ought to be used [11]. Here, the roles of managers and software developers are quintessential [23,24], as the former most often identify and stipulate an issue regarding a particular work practice, and the latter concretize a software solution that arguably affects and improves this work practice. Despite the tendency to revalue employee perspectives during software design [3,25], these intended functionalities are mostly off-the-shelve software solutions that aim to support predefined work practices in a predefined way [1,26,27,28]. In the case that employees run into problems regarding these established solutions today, they should now feel enabled to appropriate these intended functionalities ‘faithfully’ [29]; either they may do so themselves, or they may reach out to the software developer or the in-house IT support [3,13]. Second, employees may appropriate an intended functionality ‘ironically’ [11], meaning that they may use the technology in a way that is inconsistent with how it ought to be used. This may be done covertly by only using parts of an intended functionality, or by applying this functionality in ways that were not foreseen [11,16]. Third, employees may also consciously decide to cease appropriating an intended functionality.

Regarding the appropriation of unintended functionalities, employees may or may not appropriate a technology in ways that were not anticipated during implementation. In other words, new unintended functionalities may unfold in practice for purposes other than those envisioned by managers and software developers [11,16]. The general-purpose character of today’s flexible technologies especially lends itself to being exploited beyond the scope of its original intent, and may inspire employees to initiate novel functionalities that seem fitting for their existing work practices [3,13]. Empirical studies have, for instance, shown employees covertly appropriated a work-related mobile phone for making notes, for the planning of their work [30], to access workbooks, to review procedures, and to check their work-related mails [31]. Appropriation of unintended functionalities is often done covertly, and thus not made explicit. Consequently, the impact of covert appropriation on employees’ work practices and organizational improvements therefore remains largely undisclosed.

### 1.2. Continuous Improvement (CI)

Instead of starting our reasoning with technology appropriation, we could also start with the way work practices may or may not change. If we do so, something similar stands out. Employees may indeed accept work practices—defined as bounded and recurrent executions of their tasks [32,33,34]—in the way they are organized for them. However, they also may decide to deviate from these work practices. In most cases, employees that deviate from their work practices do so covertly, and thus without their manager’s knowledge or approval [35,36,37]. In sharp contrast, employees may also be stimulated by their managers to explicitly question how to conduct their work practices [17,18,19,20,38]. Making explicit that employees have the room to openly question, alter and improve their work practices continuously is the precise purpose of a continuous improvement (CI) initiative [18,39].

A CI initiative may be defined as a systematic and continuous approach in which employees are challenged to question and improve their existing work practices on a regular basis [17,19,40]. In essence, employees are expected first to actively seek issues with their existing work practices. Second, they seek and jointly discuss potential improvements to the practices. Third, they make joint decisions on how to refine their existing work practices, or even initiate novel ones. Once established, these work practices are subject to continued scrutiny as yet better work practices are always likely to be developed [17,18,40].

Interestingly, scholars studying ‘the situated use of technology’ have always highlighted organizational dynamics when it comes to explaining the contingent use of technologies [5,7,9,10,12,21]. However, the dynamics among employees within their particular context have always been limited to those dynamics that are related to technologies [5,9]. Scholars have not accounted for the sheer work practice-related dynamics fostered in an organizational context, which promotes these work practices as also being contingent [5,9]. Researching technology appropriation in such a particular organizational context may be relevant as both openly appropriating technologies, and openly discussing ones’ work practices, are actively fostered.

## 2. Materials and Methods

To document and analyze technology appropriation in an organization with a continuous improvement (CI) program in place, we conducted an in-depth case study [41,42,43,44] on how care workers appropriated a personal digital assistant (PDA) in a nursing home. Previously, empirical studies have shown that deployments of the PDA for its intended functionalities—that is to remotely register and retrieve resident data—range from great successes to utter failures [16,31,37,45,46,47,48,49,50]. The few studies explicitly elaborating PDA appropriation underline that care workers appropriate the PDA covertly, regarding both the intended [16,31] and the unintended functionalities [30,31]. For this study, however, we purposely selected a nursing home for its continuous improvement (CI) program that overtly fosters appropriation as part of a generic movement towards continuously improved work practices. In the following, we clarify (Section 2.1) the empirical setting, (Section 2.2) the data collection process, and (Section 2.3) the data analysis and synthesis.

### 2.1. Empirical Setting

In the studied nursing home, the CI program (initiated in 2015) stimulates care workers to assemble around the so-called ‘improvement boards’ on a daily basis. In particular, care workers gather around these whiteboards for fifteen minutes a day to discuss and note improvement suggestions regarding work practices and, as part of the program, regarding the distinct PDA functionalities aiming to support these work practices. As a result, care workers are stimulated to regularly reflect on problems of organizing, to highlight inefficiencies regarding existing work practices, and to initiate novel work practices [18]. In total, the nursing home employs 41 care workers—10 nurses and 31 care assistants—who care for (a maximum of) 84 residents. As these 41 potential PDA users have equally high rates of discretion and jointly question technology functionalities and work practices, the distinction between nurses and care assistants was not made in the empirical analysis.

At the start of data collection in 2017, care workers in the nursing home had already tried to appropriate the PDA for its intended functionalities. At the time, the PDA comprised an off-the-shelf software solution provided by a Belgian software developer and served as a remote assistant to support the residents’ electronic health records (EHRs). In particular, it encompassed two intended functionalities: (F1) the registration and (F2) the retrieval of resident data, such as, blood pressure, pulse and body temperature. From the moment of implementation, which was at the start of 2015, the facility manager equipped care workers during their shifts with PDAs for these two purposes. Throughout the second phase of data collection, it was noticed that care workers started appropriating the PDA for two unintended functionalities: first, as a tool to support storage and access wound pictures, and second, as a timer assistant that helped them structure their work practices. For (F3) the wound care support and (F4) the timer assistant, the PDA was appropriated in 2018 and in 2019, respectively.

### 2.2. Data Collection

We collected three sources of data: semi-structured interviews, observations and two open questionnaires. The semi-structured interviews and the observations constituted the primary dataset to examine why PDA functionalities were appropriated. The open questionnaire data were used to provide further validity. The first author conducted 27 semi-structured audio-recorded interviews of ±60 min on average: 19 with care workers, 4 with the facility manager, 2 with the head nurse and 2 with the software developer. The sequence of data collection was as follows. To start, the first author conducted a series of 14 face-to-face interviews from March 2017 to May 2017. During this first series, we covered care workers’ work practices in the nursing home and explored how the PDA was appropriated at the time. Next, a series of 13 follow-up face-to-face interviews was conducted from November 2019 to September 2020. During this second series, we focused on care workers’ appropriation of the PDA: how and why it evolved over time. Specific issues with regard to the distinct functionalities were then probed in detail. During their visits, the authors made field notes during 26 h of observations, including demonstrations on how care workers used the tool.

In addition, care workers filled out two open questionnaires. The first was distributed by the facility manager as a list of open questions in November 2019 to understand PDA appropriation (*n* = 12). In April 2020, the authors distributed a second (*n* = 32). This contained three open questions, i.e., ‘What are the distinct functionalities of the PDA?’, ‘Which of these do you use, and which not?’ and, referring to the latter, ‘Why is that the case?’ Throughout the data collection, observational data and open questionnaires were used to triangulate the interviews up until the point of data saturation. This sequence of data collection allowed in-depth knowledge to be gained about both the intended and the unintended functionalities that emerged over time (Figure 1). Table A1 and Table A2 in Appendix A provide an overview of the data collected and the semi-structured interview guide, respectively.

### 2.3. Data Analysis and Data Synthesis

All interviews, observations and open questionnaires were transcribed into text documents, which were imported into the data processing software NVivo 12. First, transcripts were examined one-by-one. Second, the first author started to code the data thematically. Third, the authors discussed their first coding and analyses to reach consensus.

To analyze the appropriation of these functionalities, insights from an inductive and a deductive qualitative approach were combined [51,52]. In particular, the three-order concept approach was applied as initiated by Gioia et al. [52]. All first-order concepts, or those that inductively covered a singular explanation for a particular functionality, were distinguished. Then, these first-order concepts were collated and thematically grouped to create second-order concepts, which related deductively to our research-centered concepts. Lastly, we deductively identified whether these research-centered concepts covered the appropriation of the distinct functionalities [51,52].

To synthesize our findings, we applied a narrative approach [53]. In doing so, below we report our findings per functionality. First, we mention how the particular work practice was executed prior to the appropriation of the related functionality. Second, we show how care workers openly discussed appropriating the technology for the coinciding work practice. Third, we mention how care workers executed the particular work practices during the last period of data collection. Interviewee quotes are used to emphasize and synthesize the data analysis [53,54].

## 3. Results

The facility manager mentioned that despite the initiatives to appropriate the PDA for its intended purposes, “*staff seem to use the PDA as they see fit. I’m not pushing them to use it, if it does not work for them*”. A care worker provided an overview of the extent to which the PDA was appropriated for intended and unintended functionalities:

*“The idea is that we raise any work-related question here. […] We are transparent about the fact that we do not use the PDA to register resident data. Some of us, however, do use the PDA to look up resident data, but actually we jointly decided to start using it all for wound care support and as a timer assistant”*—care worker.

In the following, we show how care workers openly appropriated the PDA as part of the CI program. In particular, we first show how care workers jointly decided not to appropriate the PDA for (F1) the registration of resident data. Second, we show it was decided that care workers were free to (F2) retrieve resident data as suited them best. Third, we show how care workers jointly decided to appropriate the PDA for (F3) wound care support, and (F4) as a timer assistant. For each of these individual functionalities, we analyze how care workers discussed (not) appropriating the PDA as part of the CI program to continuously improve their work practices (see Table A3 in Appendix A).

### 3.1. Registration of Resident Data

Before consulting the software developer and investing in the PDA, the facility manager and the head nurse had already been motivating the staff to accurately register resident data. Both the facility manager and the head nurse wanted the execution of this work practice to improve:

*“It is really frustrating to see that care workers use scrap paper to register: we know these notes often get lost. If they are not, one may forget to transfer them to the system. Moreover, if one does transfer them, it is inefficient because it is double work. […] Moreover, that way errors may slip into the electronic health records”*—facility manager.

In addition to problematic care efficiency, the head nurse mentioned that increasing quality of care was not met with paper-based registrations. She gave the example that once “*[…] a resident’s weight evolution had to be monitored carefully. We were embarrassed when the doctor noticed that not everything was recorded properly*”. Based on substandard registrations a software developer was sought to provide remote electronic health record (EHR) support. The PDA was introduced to make sure care workers registered resident data immediately upon measurement. The software developer acknowledged that pursuing care efficiency would require formalizing the work practice itself. Imposing restructured work practices upon care workers seemed inevitable to the software developer.

*“[H]ere, we came up against an inefficient way of working. Resident data was first collected paper-based, and only at the end of the shift entered in the system. Based on care workers’ experiences in other nursing homes, we here recommended to restructure this work practice by registering at the bedside and resident by resident, so that the electronic health records could be updated immediately upon measurement. In our opinion, installing software around a cumbersome way of working is after all pretty useless”*—software developer.

The PDA was introduced during a continuous improvement (CI) meeting during which care workers emphasized their willingness to ‘faithfully’ appropriate this intended functionality, seeing the promise to register resident data at the bedside. Compared to paper-based registrations, care workers initially agreed that registering with the PDA might reduce the loss of data and their time spent on administrative tasks. Initially, their intention to appropriate this functionality was high. However, after implementation, care workers suggested three changes that were made to appropriate the PDA for the registration of resident data on their CI meetings.

First, care workers mentioned the necessity to improve the network connection, as it urged them to continuously reconnect. In response, the facility manager had the network connection improved. In practice, however, the network connection only improved slightly. Consequently, the actual timeslots of the registrations often did not match the documented ones. The registrations were thus incorrectly delayed due to flawed network connection.

Second, at a CI meeting it was decided to purchase robust hardware as the previous PDAs were occasionally dropped on the ground. Despite the purchase, one care worker was observed being frustrated due to the difficulty in bending down, because the robust device remained a bulky one.

Third, it was openly suggested to no longer register at the bedside but in the corridor. Care workers felt that faithfully appropriating by effectively performing the registrations at the bedside would not foster but hinder other aspects of resident-centered care. At the initial stage, it was therefore decided to ‘ironically’ appropriate this functionality by registering in the corridor.

*“If I’m registering at the bedside as expected, residents look strange at me. They think I am using my personal device. They do not understand. It is also pretty time consuming that in the end I lose time I could spend with residents”*—care worker.

In total, care workers thus suggested three improvements to appropriate the PDA for the registration of resident data from its moment of implementation. However, no changes were suggested to change the off-the-shelf software solution itself. This observation is remarkable as the software solution imposed a sequential way of working, which conflicted with the convention with which care workers were familiar when conducting this work practice. Although the software developer acknowledged being open to reconfiguring the software solution after implementation in response to care workers’ improvement recommendations, he was not consulted with particular suggestions for this case. Upon consultation of the nursing home dossier, he noticed that, in 2016:

*“This nursing home only had a few user questions. These questions were actually rather generic and about various subjects. The care workers have never really pointed at particular software adjustments, they never asked for those kinds of changes. As from 2016, there have not been specific remarks or requests of this particular nursing home. From my point of view, it looks as if they are just using our software solution as intended”*—software developer.

However, during data collection in 2020, it was noticed that the care workers no longer attempted to appropriate the PDA for the registration of resident data. In particular, the software solution that required care workers to sequentially confirm resident input for every single entry resulted in confirmation fatigue. In addition, the interface of the software solution was designed so that it only allowed access to the data of one resident at a time, whereas care workers were used to having an alphabetic overview of all residents on the desktop at the ward.

*“Due to all these hurdles, I just write down the parameters of all residents on paper, the way we used to do it. At the end of my shift, I enter the data into the ward desktop. Registering resident data would mean that I have to register and confirm each parameter separately, resident by resident. What would you do if you would know that you could actually register them all at once on the desktop and confirm all parameters in one go afterwards? That works better for me. It is much more efficient and much more convenient”*—care worker.

Arguably, part of the explanation for not using this intended functionality lies in the fact this work practice became structured in an environment where care workers were used to structure their work practices themselves. Due to the directive nature of the off-the-shelf software solution, by requiring them to register one resident after the other, care workers felt they no longer had the discretion to plan other work practices. Moreover, the registration task was organized during the morning shift, when competing care activities took place. The mentioned hurdles were conceived to be more problematic at these chaotic moments, as acknowledged by the software developer:

*“Executing ten tightly scheduled tasks during a morning shift is one thing. However, making sure these ten tasks are registered adequately via the PDA may feel extra demanding during those hectic moments in the morning. Therefore, I think that time pressure is one of the biggest obstacles for these employees. Furthermore registering in itself may be at odds with what care workers actually do in their jobs. They want to be able to care”*—software developer.

As pointed out at the end of this quote, part of the explanation arguably lies in the nature of the work practice itself. The following quote underlines this aspect. Moreover, it shows that a care worker during the interview questioned why she actually had to perform the registration of resident data:

*“I just don’t have the time to register, neither with nor without the PDA. Sometimes I actually wonder why we do the registrations—I know that of course—but still, I sometimes do not see the point of it”*—care worker.

It is remarkable that care workers questioned this work practice during an interview, but not during a CI meeting. Even though the CI program was extensively applied to improve other work practices, the registration of resident data was in itself not addressed during one such a meeting.

In summary, we conclude the PDA was not appropriated for this first functionality, which was actually the key reason to invest in the PDA. We observed some improvement suggestions were made and executed to improve the hardware and the network connection after implementation. However, nothing was done about the off-the-shelf software solution. Over time, it was jointly decided at a CI meeting to stop appropriating the PDA for the registration of resident data, suggesting that care workers were able to deviate openly from using the PDA for its intended functionality: the paper-written method was renewed. Moreover, despite the flawed execution of this work practice, no other improvement suggestions were discussed. This is remarkable because it is against the nature of a CI program.

### 3.2. Retrieval of Resident Data

Based on conversations between the facility manager and the head nurse, it was decided that the PDA should “give access to all care workers so that they would have the latest updates of resident data and the care files of the residents at hand”. The purpose of the facility manager and head nurse was, on the one hand, to provide care workers with resident data (e.g., medical data and diary notes) and, on the other, to enable communication with both doctors and family members. In contrast with the spirit of the CI program, care workers themselves did not ask to improve the retrieval of resident data; they were merely informed about this second intended functionality. Nevertheless, care workers acknowledged that the retrieval of resident data may feel useful in case pharmacists, physicians or family members wanted a quick update.

*“If for instance, the family has a question or a doctor wants to know something, they have the information to hand in great detail. They can view the diary notes on it. Therefore, if someone asks how his or her mother has been the past week, care workers can open that diary and read the notes the team made about that resident. They don’t do that now, but they could!”*—head nurse.

Despite the potential to retrieve resident data easily, only some care workers in 2020 occasionally appropriated the PDA for this purpose. First, care workers did not have the tool continuously at hand, as the bulky device hindered other work practices. Second, some of the software hurdles discussed above (i.e., login hurdle, limited resident overview) also applied to retrieving resident data. As care workers had the discretion to further the previous way of working, no improvement suggestions were made to remove the hurdles.

Interestingly, some care workers perceived that using the PDA to retrieve resident data resulted in improved care efficiency compared to working without the PDA. Previously, all care workers had to tour the ward to obtain the needed information; however, it was decided at a CI meeting that care workers were able to choose how to retrieve resident data, either via the former method, or via PDA appropriation.

*“If a physician asks you whether a resident is still taking a particular medication, you can quickly request it via your PDA. So, now and then I do use it to look up information”*—care worker.

In summary, retrieving resident data is a work practice that, according to the facility manager and the head nurse, needed improvement. The software solution that intended to support this work practice was only appropriated by some care workers. Other care workers opted not to appropriate the PDA for this functionality due to technological hurdles. However, arguably due to its sporadic and unexceptional execution, improving this work practice was not discussed during a CI meeting.

### 3.3. Wound Care Support

Until the implementation of the PDA, biweekly updating the progress of wound care, e.g., storing pictures of bedsores, was an inefficient and time-consuming work practice from care workers’ point of view. At the time, the digital photo camera available at the office of the facility manager had to be held. After cleansing the wounds and before treating them, pictures of the wounds were taken. These eventually needed to be transferred manually to the right database. Nonetheless, although this practice was time-consuming and inefficient, wound care updates were previously stored regularly.

In 2018, a care worker suggested further improvement in this work practice. In particular, applying the PDA to optimize the execution of wound care support was discussed and agreed upon during a CI meeting. Soon, care workers noticed that automatically storing the wound pictures not only improved this work practice, but also the resolution of the wound pictures. From the perspective of care workers, both care efficiency and quality of care improved. Care workers found this unintended functionality of the PDA useful, as they could both in situ and remotely access previous wound pictures, store new ones, and easily track the evolution of the wound. The remote access to the evolution of the wound conceivably matched the work practice.

*“If I’m suddenly in another unit taking care of the wound of a resident I don’t normally visit, the PDA comes in handy. I can immediately access the wound care file and evaluate how the wound has evolved”*—care worker.

Moreover, one care worker emphasized that it is now easier to discuss the progress of wound pictures with a colleague. She noticed that access to the wound care files allows her now to discuss residents’ wound progress more easily with colleagues. She stipulated: “*Especially when the wound looks a bit strange, I always go and get someone. That is easier now that I have a PDA*”.

Remarkably, the PDA was appropriated for this unintended functionality, even though care workers encountered the same hurdles that hindered the intended functionalities (e.g., loose network connection, login hurdles). To support this work practice, however, these similar hurdles were not insurmountable. To the contrary, the PDA was even used in the resident’s room for this purpose, which is remarkable as the registration of resident data was explicitly not executed in the resident’s room:

*“Although the network connection improved, it is still not optimal. That is why we often stand at the window to re-enter the photographs. […] When it comes to wound care, I couldn’t do without the PDA”*—care worker.

In summary, care workers generally felt less interrupted during the execution of wound care than prior to using the PDA for the execution of this work practice. They perceived this unintended functionality both as a time- and care-efficient solution in respect to their previous way of working.

Part of the explanation for why care workers appropriated this unintended functionality may lie in the nature of the work practice itself. Irrespective of the hurdles that occurred when using the digital camera, and the hurdles that occur now that they apply the PDA, it should be noted that the wound care pictures have always been stored properly in the nursing home. Interestingly, care workers easily dealt with technological hurdles, and felt to have improved this work practice due to technology appropriation.

### 3.4. Timer Assistant

In 2017, a care worker suggested the improvement of initiating a novel work practice, i.e., washing the residents’ laundry in the nursing home to improve the quality of care. One care worker mentioned the idea of this novel work practice was mentioned at a CI meeting.

*“Well, in fact we discussed at one of these CI meetings that it might be beneficial to do the residents’ laundry ourselves in cases of bedwetting. Before, stained clothes used to remain at the resident’s room, waiting for a family member to arrive. Both colleagues and the facility manager recently agreed with this improvement suggestion”*.—care worker.

Over time, however, care workers noticed they struggled with the organization of this work practice. In particular, the washing machines provided for resident’ laundry were located in the home’s basement where care workers went independently to check the washing machines’ status. Over time, care workers realized that this work practice could be done more efficiently. Only at the beginning of 2020, however, one care worker introduced a schedule supported with a timer assistant during a CI meeting. The schedule and timer assistant made clear both who was responsible for the residents’ laundry and when to switch on or turn off the washing machines.

Care workers emphasized the usefulness and the simplicity of this second unintended functionality. One care worker mentioned: “*the timer assistant is in fact nothing but a simple alarm, but it helps to not forget to turn the washing machine back on or off*”. This simple functionality the PDA was appropriated for clearly helped care workers to structure and plan the work practice they aimed to improve. However, care workers suggested at a later CI meeting to broaden the scope of the timer assistant. During the pandemic, in particular, care workers further exploited the timer assistant by setting reminders for residents’ phone calls and video calls with their family.

*“Now that visitors are no longer allowed, we put quite a lot of effort in setting up digital meetings with the family. The timer serves as a reminder”*—care worker.

In summary, the PDA was appropriated for this second unintended functionality that enabled care workers to structure a self-initiated work practice in the first place. Secondarily, the scope of this functionality was broadened to other work practices.

## 4. Discussion

Based on a case study in a nursing home, we illustrated that care workers brought into the open novel uses of PDA functionalities, yet also overtly ceased to appropriate the PDA for its intended functionalities—both as part of their continuous improvement (CI) program. The active quest to improve their work practices arguably continuously affected technology appropriation in opposite ways. More particularly, technology was only appropriated for a work practice when employees had the discretion to both identify which work practice to improve, and how to do so. In what follows, we interpret our findings and distinguish between the appropriation of unintended and intended functionalities.

Regarding the unintended functionalities, technology appropriation was accurately embedded within the CI program [17,18,40]. Here, care workers firstly identified which work practices required improvements and addressed intrusive inefficiencies during a CI meeting. Secondly, they jointly discussed improvement suggestions to enhance these work practices’ efficiency. Thirdly, they decided how to structurally tackle these inefficiencies and perpetuate the work practice by means of PDA appropriation. In the cases of both documenting wound care and washing the residents’ laundry, care workers felt the urge to improve the respective work practice, and found a solution in the PDA to do so.

Dealing with the intended functionalities, however, was less embedded within the CI program [17,18,40]. Here, it was observed that both management and the software developer took the lead to identify and resolve work practice inefficiencies. Care workers firstly lacked the ownership to address which work practice to improve. Secondly, an alternative way of working was programmed into the software [1,55,56], and was afterwards imposed on care workers without prior consultation. After PDA introduction, we initially noticed repeated attempts to appropriate the PDA for the registration and retrieval functionalities. Later, however, care workers jointly decided not to appropriate the PDA, and to discontinue the method of registering and consulting resident data suggested by the software developer. Rejecting improvement suggestions that are not in accordance with how care workers are willing to work in practice is fully in line with the spirit of CI. However, not addressing improvement suggestions, or even rejecting the improvement of work practices that retain flaws (such as the registration of resident data), is not [17,18,20,38,40]. In essence, it is remarkable that the accustomed method of registering and consulting resident data was continued despite its flawed execution and its poor results. We therefore conclude that the varying degrees to which employees are willing to appropriate a particular functionality should be seen in a broader picture. A necessary precondition for a proper explanation of technology appropriation may be to understand employees’ perceived usefulness of a work practice itself and their willingness to improve this practice.

Although actively fostered by the CI program, care workers did not openly identify the issues they encountered regarding all of their work practices. We argue that employees’ eagerness to effectively improve a work practice may lie in the nature of the work practice itself [32,34]. Put differently, employees may deliberately choose to continuously improve one work practice, but not another. Care workers may, for instance, conceive that documenting wound pictures requires continuous improvement due to its curative nature. In contrast, the administrative nature of registering resident data may be perceived to add less value because it is less related to care workers’ meaning of proper end-of-life care. In line with this reasoning, we noticed that the documentation of wound pictures and that of resident data were executed properly and poorly, respectively—even prior to PDA appropriation. We conclude that prior to improving a work practice (via technology or not), it is quintessential for employees to see the point of their work practice, which was not the case for registration of resident data. Here, we recall a care worker quote: “*Sometimes I actually wonder why we do the registrations—I know that of course—but still, I sometimes do not see the point of it*”.

In addition to *which* work practice to improve, employees in a CI context are used to having a say in *how* they aim to improve the work practice. Based on this, we argue employees may be willing to appropriate a technology to bring structure and efficiency into one work practice, but not into another. Some work practices were discussed as they were lacking efficiency. If this was the case, the PDA was appropriated to bring efficiency into the under-structured work practices. An example is the timer assistant, which was appropriated to formalize and structure the execution of residents’ laundry. Registering resident data, however, was a work practice care workers already felt pressured by. As it was scheduled during their most hectic shift, care workers already conceived it to be too pressing, even prior to PDA implementation. Upon implementation, the software developer aimed to structure this work practice even more, which was soon conceived as over-structuring from the care workers’ viewpoint. More particularly, the software solution firstly imposed the sequence in which care workers had to visit the residents during their morning shifts, and secondly imposed the immediate storage of resident data in the EHR. Here, the imposed efficiency did not work out well and often hindered resident-centered care. We conclude that, in a CI context in which employees can take the lead in suggesting how to improve and structure their work practices, solutions from the outside may feel highly irregular.

### 4.1. Limitations and Originalities

First, as is common in qualitative case study research [41,42], not all care workers were observed and interviewed during the periods of data collection. The authors, therefore, sought to reach the remaining care workers by means of two open questionnaires. Second, the authors are aware that the case study method depends significantly on interpretation. To overcome this, various qualitative data were integrated to validate our findings. In addition, the preliminary findings were presented in the nursing home twice (Figure 1). The questions raised during the presentations for clarification did not lead to changes in the study findings.

Methodologically, this study’s research design is limited in terms of generalizing its findings to PDA appropriation in other nursing homes. However, the intention was not to generalize statistically, but to do so analytically. Analytical generalization involves scholars making assumptions about the likely transferability of study findings based on a theoretical analysis of the factors and the effect of context that both produce outcomes [41]. That is, this study’s findings may be transferred to identify why functionalities of other technologies are or are not appropriated in other contexts with high rates of employee discretion. It may also inform studies on technology appropriation in general, highlighting the willingness of employees to execute and improve their work practices.

### 4.2. Future Research Avenues

We provide four future research avenues to inform the literature on technology appropriation. First, openly providing discretion to appropriate technologies for their intended and their unintended functionalities is entirely different from explicitly providing discretion for employees to question their underlying work practices. Scholars in the area of technology appropriation, in addition to those who study technology acceptance models (TAMs), may be stimulated to account for the contingent nature of work practices. Our study shows that employees have their own view on their work practices, irrespective of whether they conduct them with a PDA. Therefore, it may be worthwhile for scholars studying technology acceptance models (TAMs) to move their discussion beyond the perceived ease of use (PEOU) and the perceived usefulness (PU) of novel technologies [57,58,59,60], by actively questioning the perceived usefulness of employees’ work practices. Moreover, a dynamic CI context may take us beyond the static task–technology fit [61], and actively account for employees’ willingness to improve their work practices.

Our second future research avenue regards the technology itself and, more particularly, the technological hurdles encountered during the execution of a work practice. We recommend scholars investigate how one technological hurdle may have different effects on the appropriation of distinct functionalities. A technological hurdle may be easily overcome in the case of one work practice, but not in the case of another. In the case of wound care support, we observed that the new technological hurdles (due to the PDA) that employees face today are easily overcome as they outbalance the inefficiencies related to the work practice prior to PDA appropriation. Gains from enhanced work practice efficiency due to PDA appropriation apparently outweighed the novel deficiencies. What constitutes whether a technological hurdle is overcome in one case and not the other is hence a crucial future research avenue.

Our third future research avenue is to more intensively study the appropriation of a technology’s unintended functionalities. Seeing the general-purpose character of novel technologies’ unintended functionalities should no longer be treated as unexpected outcomes of technology use [30,31]. Even though they have not been the purpose of implementation, it may be useful for organizations to stimulate appropriation of unintended functionalities. The ways in which organizations do or do not succeed in capitalizing on these unintended functionalities should be subject to future research.

A fourth, and final, research avenue is that scholars may benefit from more closely identifying the role of the software developer [43,62]. In our case study, the software developer acknowledged having co-created the software solution in the past with care workers from other nursing homes. Future studies may compare appropriation in such facilities, with the appropriation in facilities that the particular information system was not explicitly built for [23,63]. Regarding role of the software developer, an organization deploying a CI program may be investigated more closely par excellence. Firstly, active software developer involvement after implementation may benefit from employees who are used to reflecting on their work practices. Secondly, studies lacking interventions of a software developer in such a context may be equally relevant: it may be the case that employees feel they have the sole ownership to improve their work practices, and that the solution from the outside feels unusual for them. From the software developer’s viewpoint, the lack of questions from the nursing home arguably made him believe care workers used the PDA as intended, which was not the case. We thus recommend scholars to further investigate the alignment between software developers and employees after technology implementation.

## 5. Conclusions

The concept of technology appropriation shows employees do not only use technologies as they are expected to. Often, employees appropriate technologies differently than intended, and possibly to suit other purposes. Deviation from a technology’s intended use generally happens covertly and without the awareness and approval of employees’ managers. Our study investigated technology appropriation in an organization with a continuous improvement (CI) program in place. Such a context actively stimulates employees to deviate openly from how the work is done, in a continuous quest to seek improved work practices. Based on a case study of a personal digital assistant (PDA) in a Belgian nursing home, we make the covert end-user perspectives on the PDA overt. We find that appropriation only occurred in those cases in which care workers led the process of addressing what to improve and how to do so.

We conclude that the high rates of discretion the employees are used to (due to the CI program) arguably affect technology appropriation in opposite ways. Firstly, employees aiming to improve their work practices may continuously actively seek to broaden the scope of a technology. Secondly, employees may openly cease the appropriation of a functionality, if it is decided for them which work practice to improve and how to do so. Most remarkably, our study shows that employees who are stimulated to improve a work practice may cease doing so, despite its flawed execution.

Future scholars on technology acceptance, appropriation and use, and thus performance, should not study the technology in isolation, but in its context of use. Thus, they should explicitly take into account the perceived usefulness of the work practices the technology is intended to support. In addition, the case makes clear that a CI program fosters the dynamics of technology use. Although the program creates opportunities to bring covert appropriation processes into the open, employees’ willingness to do so is also of critical importance.

## Figures and Tables

**Figure 1 ijerph-19-06333-f001:**
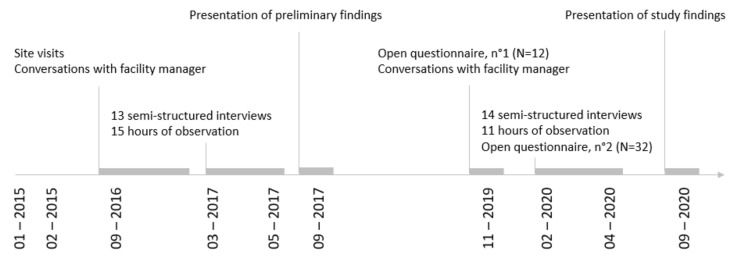
Data collection process.

## Data Availability

Data supporting reported results are available upon request to the first author.

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
