# Peer review of "‘Bringing the Covert into the Open’: A Case Study on Technology Appropriation and Continuous Improvement"

_ijerph, 2022, doi:10.3390/ijerph19106333_

Round 1
Reviewer 1 Report
The title is inappropriate while the paper does not evaluate the impact of the entire technology but specifically the use of PDA and that in a very narrow context. So, even interesting as formulation, the title does not quite correspond to the content, which is far from debating about the various existing technologies, some of them being simple to understand and to use, others being more or very complex, leading in this regard to different reactions of the people put in contact with them.
Basically, the paper analyses and concludes only on common sense things, such as that people are not robots (they don’t do what they are told to do, but they have creativity) or relativity (“We find that one task may be experienced more or less value adding by workers than another”). So, even the authors have used multiple ways to gather the data, however on a very small sample of employees (not specifying if they were more or less acquainted with the new technologies such PDA), both of these conclusions have been long ago demonstrated and accepted by people, so nothing is new.
Reviewer 2 Report
I will get straight to the point: in my opinion, the authors clearly indicate the purpose of emphasizing the exploratory nature of the research. At the same time, they themselves indicate, among others referring to publications on some of the most popular theories in this field (TAM, UTAUT), showing that there is no justification for exploring such a set goal, if not for "particular setting". And it is this "particular setting" that may be the key to the value of the article, however the authors seem to have downgraded the importance of this element in the manuscript. The authors refer to the general context of technology adaptation, however, as they themselves indicate in the introduction, the issues related to technology appropriation and work practice adaptation is not a novelty. I can only assume that the theoretical dimension and the possibility of generalizing the results were particularly important to the authors, however, it is the context of the research that may be a novelty. Therefore, one should not avoid the proper characterization of the place of implementation ("nursing home"), because the lack of it undermines the validity of the publication in Int. J. Environ. Res. Public Health. The uniqueness of this context and its meaning should be shown, and the entire story should be built on this basis. Without it, the conclusions seem less valuable: "We find that care workers in the nursing home over time did not succeed to appropriate the PDA for two intended functionalities".
There are also a lot of typos starting from line 17 and ending with 656 that suggest the need for proof reading.
Reviewer 3 Report
As a general comment, I would like to congratulate the authors for the research subject, it is a great work.
Nevertheless, there are some recommendations which I believe could improve the quality of this research work :
- in the Introduction, it might be useful to formulate the aim of the paper and to introduce research questions and hypothesis;
- in the Introduction, it might be helpful to present the structure of the article and to underline where (in which section(s)) are the own contributions presented;
- it might be important to explain the role of Section 2, in which way the literature review is connected with the developed research.
- It might be useful to verify from technical point of view the paper (for example Line 17 – t is missing);
- it might be valuable to mention why was the used software selected (Line 250); if there are other ones for these types of analysis and to present the obtained results
- it might be beneficial to present the results of data analysis, to present in which way the collected data sustained the research aims
- The paper must be completed, the mentioned Appendix in line 234 is missing.
- It might be useful to to make a more detailed presentation of the data analysis mentioned in Section 2.3, to present the obtained results by software NVivo 12-mentioned in Line 250.
- In line 247 is presented the data collection process and it might be useful to present the obtained results in concordance with it. An overview of the collected data and the way it contributed to fulfillment of the proposed research aims, can increase the understanding.
Round 2
Reviewer 1 Report
I appreciate the efforts of the authors to adjust their paper in accordance with the reviewers' observations and I remark the improvements brought by them in this regard. I still believe the subject is more suitable for another kind of journal, but as the paper looks now it may be considered also for this journal.
Author Response
Dear Reviewer
Thank you for your gentle comment:
I appreciate the efforts of the authors to adjust their paper in accordance with the reviewers' observations and I remark the improvements brought by them in this regard. I still believe the subject is more suitable for another kind of journal, but as the paper looks now it may be considered also for this journal.
Supported by your comment, we retain the repositioning of our manuscript, entitled ‘Bringing the covert into the open’: A case study on technology appropriation and continuous improvement. In addition, we executed some minor changes regarding, English language and writing style.
We once again thank you for your continued support.
Yours sincerely,
The authors

Reviewer 3 Report
Dear authors,
Thank you very much for your efforts. the article was improved. However there are some observations from my point of view:
- The paper must be completed, the mentioned Appendix in line 222 and Line 262 is missing.
- The paper must be completed, the mentioned Table A1, Table A2 and Table A3 in Lines 222 and 262 are missing
Author Response
Dear Reviewer
Thank you for your gentle comment:
R3, COMMENT 1 Thank you very much for your efforts. the article was improved.
R3, COMMENT 2 However, there are some observations from my point of view: The paper must be completed, the mentioned Appendix in line 222 and Line 262 is missing. The paper must be completed, the mentioned Table A1, Table A2 and Table A3 in Lines 222 and 262 are missing.
The missing appendices (Table A1, Table A2 and Table A3) were not included in the revised version of our manuscript.
We once again thank you for your continued support.
Yours sincerely,
The authors
